



# Impact of inflow conditions and turbine placement on the performance of offshore wind turbines exceeding 7 MW

Konstantinos Vratsinis[1,2], Rebeca Marini[1,2], Pieter-jan Daems[1,2], Lukas Pauscher[1,4,5], Jeroen van Beeck[3,1], and Jan Helsen[1,2,6]

[1]Acoustics and Vibrations Research Group (AVRG), Vrije Universiteit Brussel, Pleinlaan 2, Ixelles, Brussels, 1050, Belgium
[2]OWI-Lab, Pleinlaan 2, Brussels, 1050, Belgium
[3]The von Karman Institute for Fluid Dynamics, Waterloosesteenweg 72, Sint-Genesius-Rode, 1640, Belgium
[4]Department of Sustainable Electrical Energy, University of Kassel, Wilhelmshöher Allee 71-73, Kassel, 34121, Germany
[5]Fraunhofer Institute for Energy Economics and Energy System Technology (IEE), 34117 Kassel, Germany
[6]Flanders Make @ VUB, Pleinlaan 2, 1050 Brussels, Belgium

**Correspondence:** Konstantinos Vratsinis (konstantinos.vratsinis@vub.be)

**Abstract.**

Accurately assessing wind turbine performance in large offshore wind farms requires a nuanced understanding of how inflow parameters—turbulence intensity (TI), wind shear, and wind veer—affect power production across different turbine rows. In this study, we analyze 13 months of $10-$ minute operational data from more than 40 high-capacity turbines in a North Sea offshore wind farm, complemented by nacelle-based LiDAR measurements of inflow. Our objectives are to (1) determine how power production differs between front, middle and rear sections of the farm under the influence of TI, shear, and veer, and (2) evaluate the effectiveness of International Electrotechnical Commission (IEC)–based normalization methods, including Rotor Equivalent Wind Speed (REWS) and turbulence corrections in the front row and inside a wind farm consisting of large-scale wind turbines.

The results indicate that the impact of wind shear and veer on power output is strongly dependent on the turbine location: free stream shear and veer correlate negatively with active power in the front row, yet show positive correlations in the mid and rear rows. In addition, the TI in the wake region has a distinct influence on power production—particularly at lower wind speeds—relative to the TI observed in the free-flow region. Finally, the rear section of the wind farm exhibits approximately $20\%$ lower variability in active power relative to the front section. These location-specific changes underscore the evolving nature of inflow conditions within large wind farms. Furthermore, IEC-based REWS do not fully capture the effects of shear and veer in a large scale wind turbines in an offshore environment. The findings highlight that turbines operating in non-free-flow conditions may require additional inflow-characterization parameters beyond standard IEC norms to achieve more accurate performance evaluations and enhance overall farm efficiency.

## 1 Introduction

Global wind energy capacity continues to grow at an unprecedented rate, with new installations reaching 123 GW (World Wind Energy Association, 2024) worldwide in 2024, largely driven by robust climate policy commitments such as the EU



REPowerEU Plan (Commission, 2022) and the US Inflation Reduction Act (Congress, 2022). Although this expansion brings the international community closer to meeting ambitious decarbonization targets, it also underscores a host of technical and economic challenges. Among these are the high costs of operations and maintenance (O&M), particularly in offshore projects,
and the need to optimize the power production of turbines that are increasing in size (Ren et al., 2021).

Today, onshore wind turbines frequently exceed 4 MW in capacity, while offshore machines of 8 MW or more are becoming increasingly common (of Energy, 2024; McCoy et al., 2024; Rohrig et al., 2019; Vratsinis et al., 2024; Kelly and van der Laan, 2023). Factors such as turbulence intensity (TI), air density, wind shear, wind veer, and atmospheric stability significantly influence both their power output and structural loading (Dimitrov et al., 2015; Martin et al., 2020; Sumner and Masson, 2006).
Although larger turbines offer higher rated capacities and energy yields, their sensitivity to variations in inflow conditions can differ compared to smaller ones. For example, (Chamorro et al., 2015) demonstrated that only turbulent structures exceeding the rotor diameter can substantially affect power output, while (Van Sark et al., 2019) showed that only wind turbines with a large rotor diameter to hub-height ratio can be significantly influenced by wind shear.

Although inflow effects have been studied numerically for both free-flow and wake conditions (Saint-Drenan et al., 2020;
Sebastiani et al., 2023, 2024), most studies based on operational data have mainly focused on small to medium-sized turbines (1.5–4 MW) operating in relatively undisturbed wind conditions (Gottschall and Peinke, 2008; Wagner et al., 2009; Murphy et al., 2020; Clifton and Wagner, 2014; Bardal and Sætran, 2017; Kim et al., 2021; Mata et al., 2024; Gao et al., 2021; Wagner et al., 2011). Many studies explore the effect of wake on power production from the perspective of velocity deficit (Adaramola and Krogstad, 2011; González-Longatt et al., 2012), but relatively few investigate the power performance of
wind turbines within wind farms under waked conditions. Consequently, while turbines inside wind farms often experience inflow conditions that deviate significantly from free-stream flow, the impact of these deviations on power production remains relatively unexplored.

Industry-standard normalization procedures for evaluating wind turbine performance are outlined by the International Electrotechnical Commission (IEC) (IEC, 2022). These methods correct for environmental factors such as TI, air density, wind
shear, and veer, and are widely used to compare the performance of different turbines under various conditions. However, they were developed primarily with single, isolated turbines in mind, and their applicability to the wake-affected regions of large wind farms is not recommended and has not yet been explored. Indeed, recent work suggests that standard IEC-based turbulence corrections can both overcompensate and undercompensate for inflow TI, potentially leading to inaccuracies in power curve estimations (Lee et al., 2020).
Motivated by the growing size of offshore wind projects and the scarcity of studies based on large-scale offshore wind turbines, this study uses extensive real-world operational data from a large offshore wind farm to address two main questions. First, we investigate how inflow parameters and specifically TI, wind shear, and veer impact power production across different segments of an offshore wind farm that includes wind turbines exceeding 7 MW rated power. Second, we answer to what extent do the established IEC-based normalization and correction methods mitigate these inflow effects in free-stream and highly
wake-affected segments of the wind farm. By examining real-world SCADA data and applying industry-standard corrections, the study aims to clarify whether current IEC methods remain robust for modern wind turbines or require further refinement,





and it also explores the possibility of applying these methods across an entire wind farm. Ultimately, these insights are intended to improve the accuracy of large offshore wind farm performance assessments and help reduce uncertainties in both technical design and financial planning.

The paper is organized as follows. Section 2 describes the datasets and filtering methods. Section 3 outlines the methodology, including the selection of specific sections of the wind farm, the corrections implemented, the correlation analyses conducted, and the variability analysis. Section 4 presents our results, examining each of the three corrections individually. Finally, Section 5 summarizes and discusses the key findings.

## 2   Data collection and filtering

The study uses $13$ months of data from an offshore wind farm located within the Belgian maritime area. This wind farm comprises over $40$ turbines positioned more than $30$ kilometers from the coastline. The prevailing wind direction is from the southwest, which predominantly influences the performance of the wind farm.

For the analysis, two different datasets are used:

**SCADA data**: Collected from the wind turbines, providing real-time operational information. This includes wind speed,
active power, turbulence intensity (TI), and direction measured by the SCADA system using a combination of cup anemometers, wind vanes, and sonic anemometers mounted on each turbine's nacelle.

**LiDAR data**: Measured at a nearby wind farm approximately $8 km$ away (see figure 1). A continuous-wave nacelle-based LiDAR (ZX TM) is used to estimate wind shear and veer. The LiDAR data are averaged over $30-min$ periods for calculating wind shear and veer. Given the position of the LiDAR relative to the selected wind sector, we assume
that the estimated shear and veer are representative of the entire wind farm. Therefore, the travel time to reach different sections of the wind farm has not been considered during this analysis.

### 2.1   Data filtering

Several filtering steps are applied to the combined dataset to ensure data validity:

1. **Data availability**: Data from $1-sec$ measurements are aggregated into $10-min$ averages, allowing up to $100$ seconds
of missing data per $10-min$ window. This tolerance increased data availability without significantly impacting the uncertainty of the aggregated values.

2. **Turbine operational regime**: Only data from operational turbines, as indicated by SCADA operational status codes, are included.

3. **Data sanity**:

**Low variability**:$10-min$ intervals where the standard deviation of wind speed or active power was less than $0.01\%$ of its mean value are removed to eliminate erroneous data points marked as rejected data.





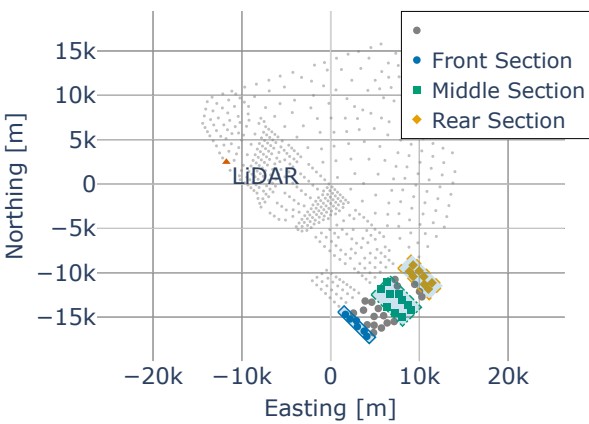

**Figure 1.** Wind farm layout showing the LiDAR system and the designated study sections for southwest wind conditions. Gray dots represent other wind farms within the same cluster.

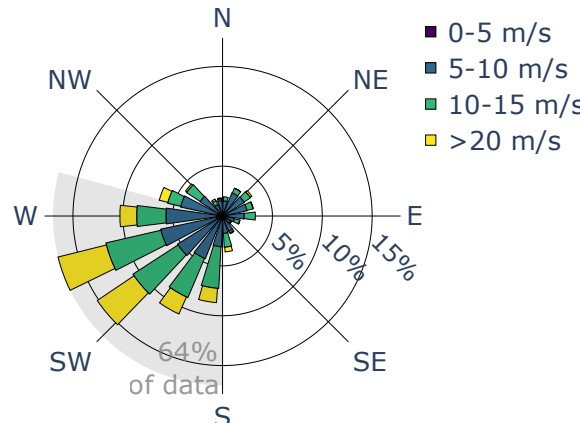

**Figure 2.** Wind rose illustrating the distribution of wind directions and rotor wind speeds, highlighting the study region where 50% of the data points are concentrated.

**Outliers**: Outliers with respect to the power curve were identified and excluded from the dataset using the DB-SCAN (Density-Based Spatial Clustering of Applications with Noise) algorithm, utilizing wind speed and active power as the two feature dimensions for clustering.

4. **Wind sector selection**: The analysis focused on the wind sector ranging from $180°$ to $285°$, which accounted for approximately 50% of the observations (see Fig. 2) during the study period. This specific sector was chosen to ensure that the LiDAR is measured upwind of the wind farm and is free from wake effects. This selection allows for accurate estimation of wind shear and veer.

After these filtering steps, the final dataset comprised over $600,000$ $10-min$ intervals, representing approximately 25% of the raw data for the studied wind sector.

## 3 Methodology

This study evaluates the correlation of environmental parameters across different sections of a wind farm. It also assesses the potential application of IEC corrections in SCADA-based performance evaluations within an offshore wind farm, specifically by examining their impact on correlations with power production and the variance of the power curve. To achieve the objectives of the study, after the collection and filtering of the data, the following procedure is followed:

**Section division**: The dataset is divided into three sections: the front (first row with free-stream inflow), the mid (middle section), and the rear (end of the farm).

**Active power normalization**: Power production is normalized using the power curve of the manufacturer .



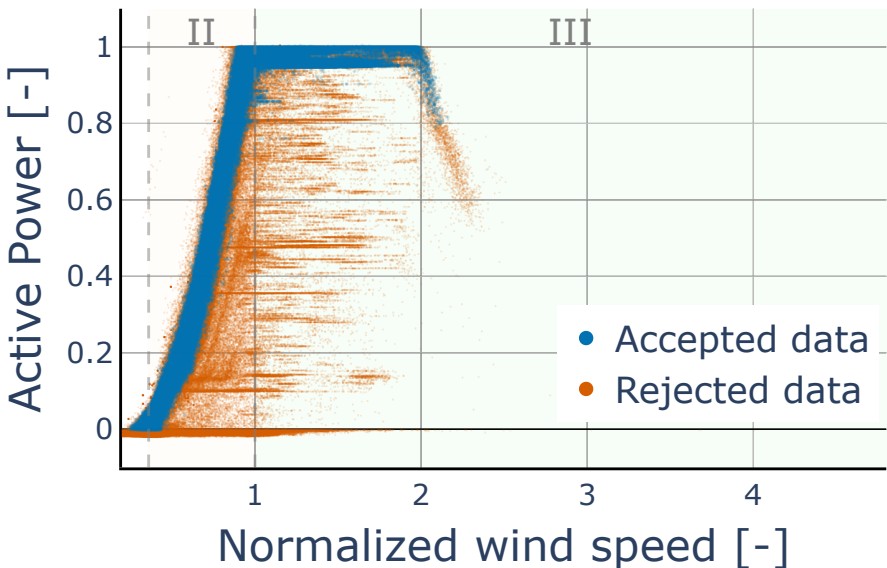

**Figure 3.** The raw dataset is shown: green indicates the accepted data used in this study, and red denotes the rejected data. The region II is referring to torque control region and region III to pitch control

**IEC corrections**: The wind speed and the power output are adjusted using the IEC corrections IEC (2022) for wind shear-veer and TI.

**Correlation analysis**: Correlation analysis is performed between environmental parameters and active power before and after the IEC corrections for each section. The effect of IEC corrections is then evaluated.

**Variance analysis**: The variance of the active power is traced over different wind speeds, both before and after the IEC corrections, for each section. The effect of IEC corrections on the power variability is then evaluated.

These steps are further explained in the following sections of this paper.

## 3.1 Section division

Free-flow wind turbines are defined in accordance with the IEC 61400-12-1:2022 IEC (2022) standard, which specifies the criteria for an undisturbed region by accounting for the influence of adjacent wind turbines and obstacles. We apply the recommended method to calculate the valid sectors for each wind turbine in the wind farm, selecting those in the first row that met the standards as free-flow turbines. The middle and rear sections are defined based on the distance to the nearest free-stream wind turbine measured along the flow direction. The sections widths are defined so as to ensure they contain a volume of data





comparable to that from free-flow wind turbines—approximately $160,000\ 10 - min$ intervals per section. Hence, the datasets are balanced and easily comparable. Figure 1 illustrates the segmentation of the wind farm into these three sections.

## 3.2 Active power deviation

This study examines how each environmental parameter positively or negatively contributes to power output at various locations within a wind farm and across different wind speeds. To isolate the influence of these environmental variables while minimizing the effect of variations in wind speed within each bin, we employ the concept of power deviation (PD) to normalize the active power, as described below:

$$PD_i = P_{\text{measured},i} - P_{\text{manufacturer},i} \tag{1}$$

where $P_{\text{measured},i}$ refers to the measured power at the $i^{th}$ wind speed bin, and $P_{\text{manufacturer},i}$ refers to the power given by the power curve of the manufacturer at the $i^{th}$ wind speed bin.

## 3.3 IEC corrections

The atmospheric boundary layer in which a wind turbine operates is a dynamic environment. Different inflow conditions are expected to produce varying power outputs at a wind speed. However, collecting sufficient data in a short time frame for
every possible inflow condition to create a multi-variable power curve is practically impossible. The IEC standards recommend a combination of corrections to wind speed and active power to reduce the dependency of power output on environmental parameters. One of the goals of this study is to assess the correlation between each environmental parameter and power output across different sections of the wind farm and evaluate the effectiveness of existing IEC corrections in these sections. Ideally, after applying the corrections, the correlation between the environmental parameters and power output should approach zero.
This section will briefly discuss the corrections applied in this study.

### 3.3.1 Wind shear and veer correction

For wind turbines with large rotor diameters, variability in wind speed and direction across the rotor can significantly affect power production. This study employs LiDAR measurements taken upstream near the wind farm to evaluate the shear and veer coefficients of the wind. The derived coefficients enable estimation of wind speed and direction at different heights based on
the average shear and veer profiles.

According to IEC standards (IEC, 2022), the REWS is defined as:

$$v_{\text{rews}} = \left( \sum_{i=1}^{n} \left( v_i \cos(\phi_i) \right)^3 \frac{A_i}{A} \right)^{1/3}, \tag{2}$$

where $n$ is the number of available measurement heights; $v_i$ is the wind speed calculated at height $i$ based on the shear exponent and the hub height wind speed; $\phi_i$ is the calculated angle difference between the rotor direction and the wind speed at height $i$
based on the measured difference at hub height; $A$ is the swept rotor area; $A_i$ is the $i^{th}$ segment area.



To assess the individual contributions of shear and veer to the REWS, we compute the REWS under two separate conditions: one that excludes veer ($\phi = 0$) and one that excludes shear ($u_i = u_{hubheight}$). The resulting REWS values are then applied to normalize the wind speed according to the IEC standard.

### 3.3.2 Turbulence Intensity (TI) correction

TI can significantly impact the power output of a wind turbine, particularly within a wind farm, where TI tends to increase (Barthelmie et al., 2007), potentially biasing the power curves across different sections. This study employs the TI normalization recommended by the IEC to model the effects of $10 - min$ averaging on power output.

A complete discussion of the normalization process is outside the scope of this paper and is well documented in the IEC standard (IEC, 2022); here, we provide an overview for completeness. The first step is to calculate the zero turbulence power 155 curve. The zero turbulence power curve represents the theoretical power output of a wind turbine under idealized conditions in which the wind is completely steady. The normalization process adjusts the active power measured during a 10-minute interval by first subtracting a simulated average power—calculated using the ideal zero TI power curve and the measured wind distribution—and then adding a simulated average power—calculated using the ideal zero TI power curve and a Gaussian wind speed distribution corresponding to the reference TI. The normalization is given by:

$$\overline{P_{I_{\text{ref}}}(v)} = \overline{P(v)} - \overline{P_{\text{sim},I}(v)} + \overline{P_{\text{sim},I_{\text{ref}}}(v)}, \tag{3}$$

where, $P_{I_{\text{ref}}}(v)$ is the normalized power output; $\overline{P(v)}$ is the measured power output; $\overline{P_{\text{sim},I}(v)}$ is the simulated average power output based on the real wind distribution using the zero TI power curve, and $\overline{P_{\text{sim},I_{\text{ref}}}(v)}$ is the simulated average power output at the $I_{\text{ref}}$ based on the Gaussian wind speed distribution using the zero TI power curve.

### 3.4 Correlation and linear regression slope analysis

The correlation analysis serves a dual purpose. First, it aims to elucidate the relationship between power production and inflow conditions at different locations within the wind farm. Second, it evaluates the effectiveness of IEC corrections within the wind farm, which falls outside the current scope of the standard. To achieve these objectives, a Sliding Window Pearson Correlation (SWC) approach was employed. Because the control state of a wind turbine is highly dependent on wind speed during normal operation, the data are first sorted by wind speed, and the sliding window is applied along this dimension. 170 Pairwise correlations are calculated within a sliding window with a width of $0.75 \text{ m s}^{-1}$. The correlations obtained from each window are then plotted against the mean wind speed of that window, which allow the identification of how the relationship between environmental variables and power production shifts across different wind speeds, both before and after the correction. Similarly to the correlation analysis, we computed the linear regression slope of TI with respect to the normalized active power deviation, defined as $PD_{\text{norm}} = \frac{PD}{\text{rated power}}$, where $PD$ is the active power deviation. This slope provides another measure to 175 assess the performance of the TI correction. By plotting the slope against the mean wind speed for each window, both before and after the correction, we can observe how the sensitivity of active power to TI changes with wind speed. This approach





thereby complements the insights gained from the correlation analysis by evaluating the effectiveness of the correction in reducing the influence of TI on power output.

### 3.5 Power curve variability analysis

While correlation analysis is useful for examining whether an environmental parameter influences power output before and after a correction, it has its limitations. Applying a correction can sometimes introduce noise that reduces the observed correlation between the environmental parameter and active power, effectively masking the true dependency. This means that relying solely on correlation analysis may not fully capture the impact of the correction. To more effectively assess the effect of a correction, we propose examining the variability of active power at different wind speeds across various sections of the wind farm, both

before and after the correction. Specifically, we use the Median Absolute Deviation (MAD), as shown in Equation (4). This approach allows us to identify how the corrections affect the variance of the power curve. Moreover, MAD is less sensitive to outliers, making it a robust measure for this analysis.

$$\mathrm{MAD}_{P_j} = \mathrm{median}\left(\left|P_{j,i} - \mathrm{median}(P_j)\right|\right) \tag{4}$$

where $P_{j,i}$ is the active power measurement $i$ at wind speed bin $j$; and $\mathrm{median}(P_j)$ is the median of all active power measure-

ments at wind speed bin $j$.

## 4   Results and discussion

This section analyzes the influence of environmental parameters—wind shear, veer, and TI—on power production in different sections of the wind farm, both before and after applying IEC corrections. Three main types of plots are utilized: (1) Pearson correlation plots that show the relationship between each environmental parameter and active power across wind speed win-

dows; (2) Corresponding Pearson correlation plots for the corrected values, which allow for an assessment of the effectiveness of the corrections; and (3) for TI, additional linear regression slope plots that evaluate the sensitivity of active power to TI before and after the correction.

### 4.1 Wind shear and veer correction

Figures 4 and  5 illustrate the correlation of wind shear with active power before and after applying the REWS correction. A key

observation is that the correlation patterns between wind shear and power production change depending on the position within the wind farm. Specifically, the influence of the free-flow incoming wind shear is most pronounced in the front section, with its influence weakening further downstream. Negative correlations are observed in the front row, while positive correlations appear in the rear section. This shift may be explained by changes of the free-flow wind profile as it moves through the farm, where wind turbine wakes enhance mixing and alter the shear and veer characteristics, however further investigation is required

to understand the mechanism.





Figure 5 shows that even though the REWS correction is applied based on the time-averaged inflow profile, it fails to reduce the correlation with the wind shear on the front section or within the wind farm. This could be a result of the small correction that the REWS applies to the wind speed as suggested on (Van Sark et al., 2019). In our case, based on the available measurements, the average correction is approx. $0.1 \text{ m s}^{-1}$, with a standard deviation of $0.08 \text{ m s}^{-1}$

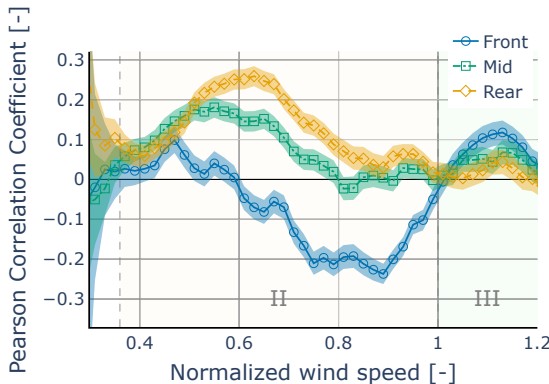

**Figure 4.** (a) Correlation between wind shear and active power output at various wind speeds before applying the REWS normalization.

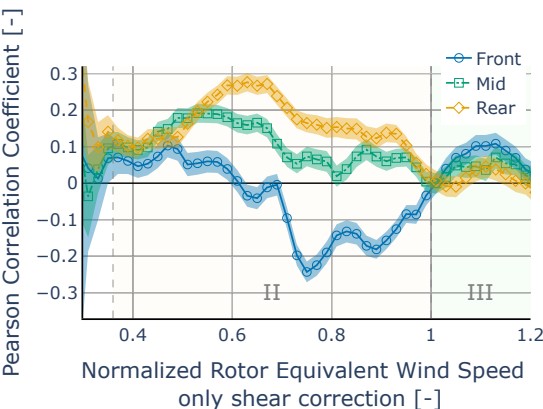

**Figure 5.** Correlation between wind shear and active power output at various wind speeds after applying the REWS normalization only for shear.

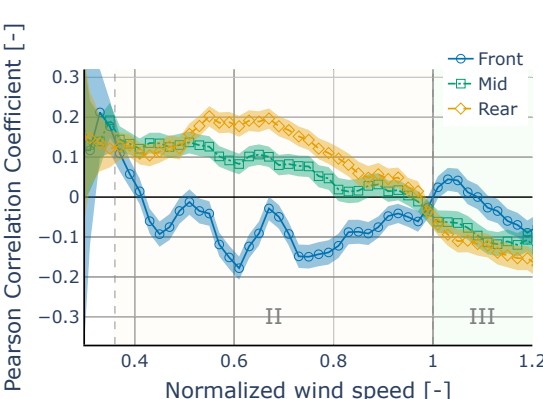

**Figure 6.** Correlation between wind veer and active power output at various wind speeds before applying the REWS normalization only for veer.

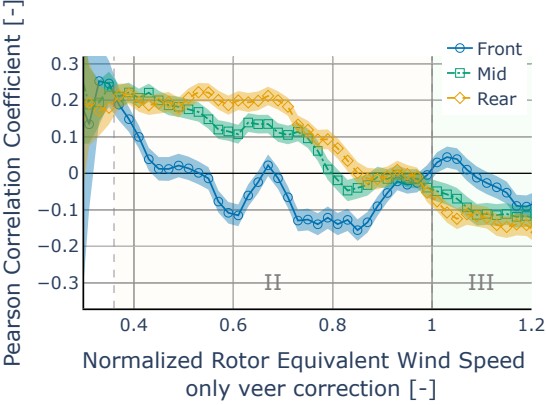

**Figure 7.** Correlation between wind veer and the power deviation at various wind speeds after applying the REWS normalization only for veer.

Although the distance to measure wind shear and veer is relatively short for an offshore setting, obtaining more precise data closer to the wind farm - without assuming a wind shear profile based on power law Debnath et al. (2021) - may improve the





correction. Another possible explanation is that shear and veer are correlated with other factors, such as turbulence intensity, which also affect power output. This interdependence makes it challenging to correct using REWS alone.

Similarly, the correlation patterns between wind veer and power output change depending on the position within the wind farm. In the front section, wind veer is negatively correlated with turbine power output, while positive correlations are observed in the rear section. Given that veer is linked with shear Kelly and van der Laan (2023), our results suggest that both parameters exert a similar spatially dependent correlation with turbine performance. The REWS correction has a small impact on the effect of wind veer in the front section but does not decouple the correlation of inflow of free stream wind veer from the power output of the wind turbine. It is important to note that the REWS correction is primarily designed to adjust for variations in the energy flux over the rotor due to wind shear and veer. However, the wind profile can also affect the aerodynamic load on the turbine blades. Consequently, even after applying the REWS correction, the residual correlation between wind veer or shear and power output indicates that these effects may not fully mitigated.

## 4.2 TI correction

Figures 8 and 9 compare the correlation between TI and wind turbine power output before and after applying the IEC-based correction at different locations within the wind farm. Additionally, figures 10 and 11 illustrate the sensitivity of the active power to TI by presenting the slope $\beta$ of the linear regression between TI and normalized PD. In figure 8, the correlation of TI at various wind speeds aligns with the expected behavior based on theoretical modeling (Saint-Drenan et al., 2020). Specifically, as expected by the literature, the front section has a positive correlation for normalized wind speeds 0.36–0.84 and a negative correlation at higher wind speeds. However, the rear sections have significantly different behavior for normalized wind speeds of up to 0.84. Although the negative correlations are initially small at wind speeds below 0.7, they become considerably stronger as the wind speeds approach the rated value. All sections have similar behavior at normalized wind speeds greater than 0.84; however, there are significant differences at lower wind speeds. Although all sections exhibit similar behavior at normalized wind speeds greater than 0.84, the significant differences at lower wind speeds suggest that the turbulent characteristics in the free-flow region are significantly different from those in the waked region.

The slope analysis presented in figure 10 complements the correlation findings by quantifying the sensitivity of PD to TI. The slopes $\beta$ indicate how much the PD changes with a unit change in TI. The front section shows a small positive slope in the normalized wind speed range of 0.4 to 0.65, indicating a positive effect of TI on power. In contrast, in the same wind speed ranges, the mid and rear sections show lower slopes, indicating that these sections of the wind farm have a lower sensitivity to TI compared to the front section at low wind speeds. However, for normalized wind speeds above 0.84, all sections show a higher sensitivity to TI.

The results of the IEC correction in figure 9 indicate that, in the front and mid sections, the IEC correction reduces the correlation by about half for normalized wind speeds between 0.4 and 0.6, and further decreases TI dependency for wind speeds greater than 0.8. However, the correction behaves differently in the rear sections, where it overcompensates for TI effects and even shifts the correlation to the negative side for a wind speed lower than 0.6. For normalized wind speeds above 0.84, the correction appears to eliminate most of the TI dependency.



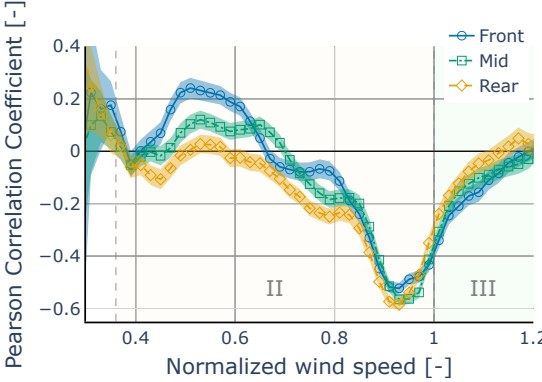

**Figure 8.** Correlation between TI and active power output at various wind speeds before applying the TI normalization.

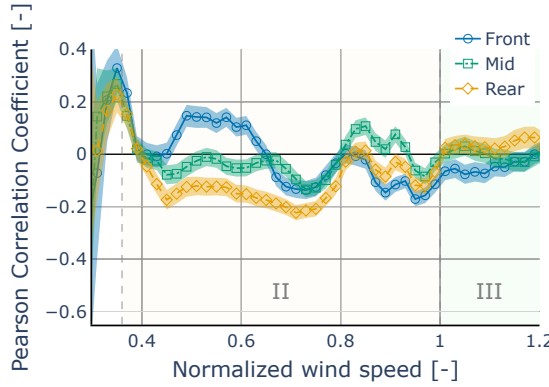

**Figure 9.** Correlation between TI and active power output at various wind speeds after applying the TI normalization.

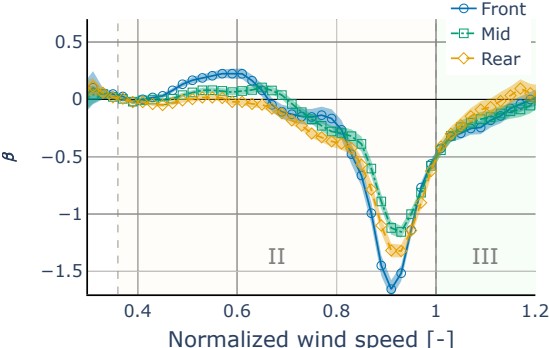

**Figure 10.** Linear regression slope between TI and active power output at various wind speeds before applying the TI normalization.

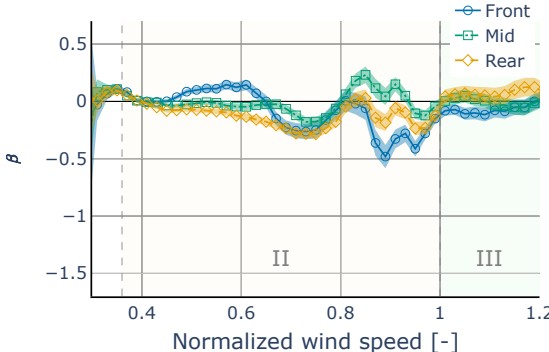

**Figure 11.** Linear regression slope between TI and active power output at various wind speeds after applying the TI normalization.

Similarly, the slope analysis in figure 11 shows that the sensitivity of PD to TI is significantly reduced after the correction in all sections for high wind speeds. For low wind speeds in the rear section, the slopes become more negative or remain low, indicating that the correction may be overcompensating or not adequately accounting for wake-induced turbulence effects. The overcompensation observed in the rear sections may be due to the increased wake-induced turbulence, which might differ significantly from the free-stream turbulence conditions assumed in the IEC correction methodology.

This suggests that the IEC TI correction may not be fully applicable within the wake-affected regions of the wind farm. However, it can still significantly correct for a large part of the TI effect on power production.





## 4.3 Effect on the variability of the power curve

To further examine how turbine location affects power production, we quantified the variability of active power in each section
using the MAD. First, we established a baseline for each section (Figure 12) representing the variability before any correc-
tions. In contrast, the rear section exhibits less variability than the front section, despite the presence of wakes and increased
turbulence inside the wind farm.

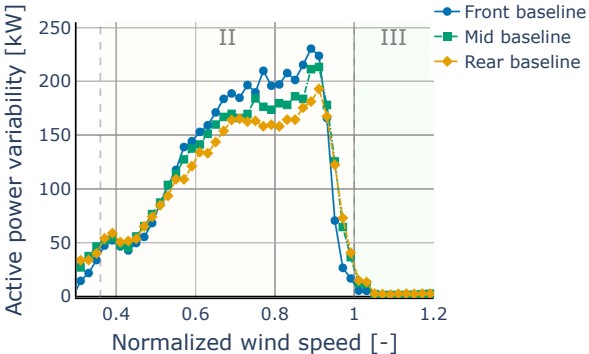

**Figure 12.** MAD of active power output across different wind
speed bins for the different sections of the wind farm. The base-
line curve represents the power curve MAD without any correc-
tions.

**Figure 13.** Percentage change in the MAD of active power out-
put for the mid and rear sections compared to the front section.

For clarity, figure 13 shows the percentage change in MAD between the mid and rear sections compared to the front section.
Both sections show reduced active power variability for normalized wind speeds between $0.64$ and $0.92$, with the rear section
showing reductions of up to $40\%$. Next, we evaluated how the applied corrections influence this variability. Figure 14 illustrates
the changes in MAD after each correction for each section. Overall, the corrections result in relatively small changes in MAD,
with the only significant reduction in active power variability occurring from the TI correction for below-rated conditions.

## 4.4 Evaluation of correction-induced correlation changes

To visualize how each correction affects the correlation between environmental factors and power output, figures 15, 16, and
17 present the absolute change in correlation in the front, mid, and rear sections of the wind farm. The metric shown is
$(|\rho_{\text{after correction}}| - |\rho_{\text{before correction}}|)$ where $\rho$ is the Pearson correlation coefficient, as a function of wind speed, with different
lines representing the corrections for wind shear, veer, and TI.

As shown in these figures, the REWS corrections for wind shear and veer offer minimal reductions in correlation within the
front section. In the mid and rear sections, these corrections may even slightly increase the correlation at lower wind speeds.
In comparison, the effect of the TI correction is much stronger. The TI correction has a smaller effect in the front section, but it



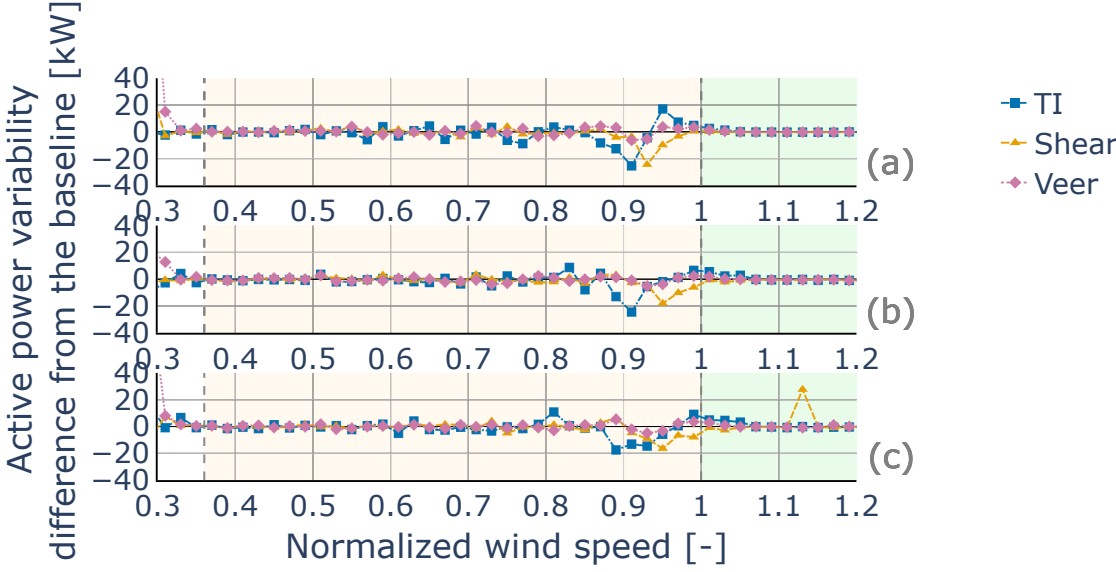

**Figure 14.** Differences in the MAD of active power output after each correction from the baseline across different wind speed bins. (a) Differences in the front section, (b) in the mid section, and (c) in the rear section of the wind farm.

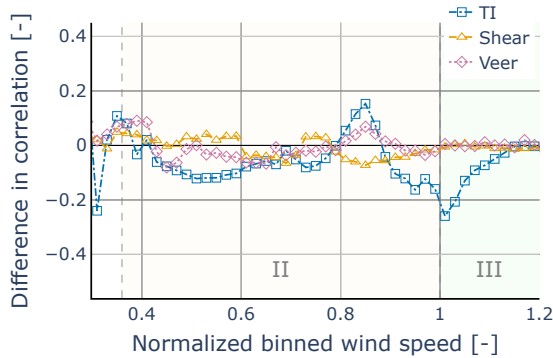

**Figure 15.** Change in the magnitude of the correlation between environmental factors and power output before and after applying corrections, plotted against wind speed for the front section of the wind farm.

becomes more significant in the mid and rear sections, particularly as wind speeds approach the rated value, while the picture is not as clear for lower wind speeds.





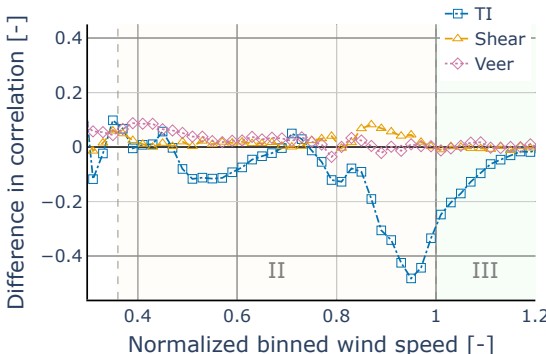

**Figure 16.** Change in the magnitude of the correlation between environmental factors and power output before and after applying corrections, plotted against wind speed for the mid section of the wind farm.

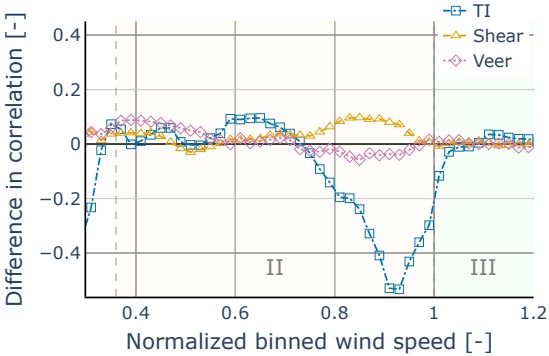

**Figure 17.** Change in the magnitude of the correlation between environmental factors and power output before and after applying corrections, plotted against wind speed for the rear section of the wind farm.

## 5 Conclusion

Using $10-$minute SCADA data, we evaluated the dependence of power production on turbulence intensity (TI), wind shear, and wind veer across different sections of a large offshore wind farm. Our findings demonstrate that these environmental factors influence turbine power output differently based on their location within the farm. Specifically, we highlight the limitations of IEC-based corrections, namely the TI correction and the rotor equivalent wind speed (REWS), by comparing their effectiveness between free-stream turbines (front section) and wake-affected (mid and rear sections). **Wind Shear and Veer**: Our analysis reveals a clear negative correlation between wind shear and veer with power production in the front-section turbines, aligning with expectations for free-stream conditions. In contrast, turbines in the mid and rear sections exhibit a positive correlation, suggesting that higher shear and veer may enhance power production within the wind farm. This behavior is likely due to the redistribution of energy by front-section turbines, which reduces shear and veer downstream. However, the absence of simultaneous wind shear measurements within the farm and the considerable distance of the nacelle LiDAR from the turbines limit our ability to fully explain this mechanism. Additionally, despite the large rotor sizes, the IEC-based REWS correction does not effectively model the relationship between shear, veer, and active power, showing minimal impact on altering the correlation between wind speed and active power across all farm sections. **Turbulence Intensity (TI)**: Our correlation analysis indicates that TI positively correlates with active power when data from the front and mid sections are analyzed at low wind speeds ($< 0.8$ of the rated value), which is consistent with findings reported in the existing literature. However, in the rear section, TI exhibits a negative correlation with active power, highlighting that wake-induced TI affects power production differently compared to free-stream TI. Furthermore, the IEC-based TI correction successfully models the impact of TI at lower wind speeds for the front and mid sections but fails to do so for the rear section. At higher wind speeds ($> 0.8$ of rated),



the IEC-based TI correction is effective across all sections. In general, under the free flow conditions considered by the IEC, the TI correction performs as expected. **Overall Power Production Variability**: Our study demonstrates that the variability of power production is lower in the mid and rear sections compared to the front section, reflecting spatial differences in the flow
conditions within the wind farm that are not fully captured by the available measurements. This reduced variability suggests that inflow conditions evolve significantly as the wind passes the farm, leading to more stable power output in the downstream turbines.

These findings underscore that turbines operating under non-free-flow conditions within a wind farm experience inflow characteristics that differ substantially from free-stream scenarios. Consequently, current IEC-based normalization methods are
insufficient for accurately correcting the effect of TI, shear and veer in wake-affected environments. To improve performance evaluations, it is essential to incorporate a wider range of inflow characteristics, such as turbulent kinetic energy (TKE) (Kumer et al., 2016), beyond standard IEC parameters. This includes a more detailed characterization of wake-induced turbulence and shear effects, which can enhance the accuracy of power curve predictions and reduce uncertainties in the technical design of large offshore wind farms.

Overall, our study underscores the limitations of current flow characterization methods to accurately assess power performance in large-scale offshore wind farms. Although the TI correction for free flow wind turbines at wind speeds around the rated performs as expected, the results for the mid- and rear sections are mixed. This highlights the need for improved normalization and correction techniques tailored to the complex inflow conditions encountered within such environments. By addressing these challenges, the wind industry can more effectively optimize turbine performance, reduce operational costs
through improved power curve estimations, and achieve greater reliability in meeting global renewable energy targets.

## 6 Code/Data Availability

The data supporting the findings of this study are not publicly available due to confidentiality restrictions under non-disclosure agreements with the data provider.

## 7 Author contributions

KV undertook the tasks of data curation and formal analysis, as well as writing, reviewing, and editing the initial draft and the final manuscript. RM was responsible for analyzing the LiDAR data. PJD, LP, JvB, and JH provided supervision, validated the results, and contributed to the review and editing of the manuscript. Finally, JH secured the necessary funding for this work.

## 8 Competing interests

The authors declare that they have no conflict of interest.





# 9 Acknowledgements

The authors acknowledge the financial support via the MaDurOS program from VLAIO (Flemish Agency for Innovation and Entrepreneurship) and SIM (Strategic Initiative Materials) through SBO project Rainbow. The authors would moreover like to acknowledge the Energy Transition Funds of the Belgian Federal Government for their funding of the POSEIDON project. Finally, the authors acknowledge the support of De Blauwe Cluster through the project Supersized 5.0. This work used large language model software for spelling and grammar checks.



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
