# Peer review of "Impact of inflow conditions and turbine placement on the performance of offshore wind turbines exceeding 7 MW"

_Wind Energy Science, 2025_

## Referee Comment (RC1)

Review of the paper "Impact of inflow conditions and turbine placement on the performance of offshore wind turbines exceeding 7 MW".

Submitted to *Wind Energy Science* Article number #: wes-2025-32 March 25, 2025

**Recommendation**: Minor revision. Summary:**

This paper provides an insightful and detailed analysis of wind turbine performance in a large offshore wind farm, focusing on the influence of inflow parameters such as turbulence intensity (TI), wind shear, and wind veer on power production across different turbine rows. The topic is of considerable interest, and the methodology is robust. This paper makes a contributions to the field of wind energy by detailing how specific atmospheric conditions affect power generation in different sections of a wind farm and challenging the adequacy of current standard measures. However, several issues must be addressed before the manuscript can be recommended for publication. My comments are categorized as either 'Major concerns' or 'Minor concerns', with the former focusing on conceptual technical critiques, and the latter highlighting grammatical and spelling errors.

**Major concerns:**

- (1): In the manuscript, on page 7, line 160, there is a mention of " $P_sim$ ", which refers to the simulated average power output. However, the document does not provide any information on how this simulated power is calculated. For clarity and to maintain the integrity of the study's methodology, it is essential to include a detailed description of the process used to calculate the simulated power output.
- (2): Figure 1. "Wind farmlayout showing the LiDARsystem and the designated study sections for southwest wind conditions. Gray dots represent other wind farms within the same cluster.". However, the figure includes two shades of gray dots—light and dark—and there is no explanation provided in the figure caption or the accompanying text regarding their distinct meanings.
- (3): Figure 3 "The raw dataset is shown: green indicates the accepted data used in this study, and red denotes the rejected data. The region II is referring to torque control region and region III to pitch control". However, the actual figure does not contain these colors, which could lead to confusion for readers attempting to interpret the data.
- (4): Figures 15, 16, and 17 mention "...before and after applying corrections...". However, the figures currently only display three lines representing TI, shear, and veer without clearly differentiating the data before and after the application of corrections.
- (5): It needs to clearly define how "free-stream TI" and "free stream TI" are calculated.
- (6): In the conclusion section, it mentions "our study underscores the limitations of current flow characterization methods to accurately assess power performance in large-scale offshore wind farms", however, there is limited discussion on "current flow characterization methods" in the introduction section.

References

---

## Author Comment (AC1)

**Response to Reviewers**

*Manuscript: Impact of inflow conditions and turbine placement on the performance of offshore wind turbines exceeding 7 MW*

*Manuscript ID:* WES-2025-32    *Journal:* Wind Energy Science

*Corresponding author:* Konstantinos Vratsinis

The authors thank the editor and the reviewers for their thoughtful and constructive feedback. We revised the manuscript accordingly. In the following, we reproduce each comment in full and provide a point-by-point response, indicating where changes appear in the revised manuscript. A brief note titled **Global revision due to the representativeness update** appears at the end of this document and summarizes the global figure and data updates. In particular:

**Reviewer 1**

"This paper provides an insightful and detailed analysis of wind turbine performance in a large offshore wind farm, focusing on the influence of inflow parameters such as turbulence intensity (TI), wind shear, and wind veer on power production across different turbine rows. The topic is of considerable interest, and the methodology is robust. This paper makes a contributions to the field of wind energy by detailing how specific atmospheric conditions affect power generation in different sections of a wind farm and challenging the adequacy of current standard measures. However, several issues must be addressed before the manuscript can be recommended for publication. My comments are categorized as either 'Major concerns' or 'Minor concerns', with the former focusing on conceptual technical critiques, and the latter highlighting grammatical and spelling errors. " Thank you for your detailed comments, which helped us to significantly improved the clarity and consistency of our figures and methodological descriptions.

**Comment 1:**

In the manuscript, on page 7, line 160, there is a mention of "$P_{sim}$", which refers to the simulated average power output. However, the document does not provide any information on how this simulated power is calculated. For clarity and to maintain the integrity of the study methodology, it is essential to include a detailed description of the process used to calculate the simulated power output.

**Response:**

We added a definition of $P_{sim}$ and minimally reform part of the section to clarify that.

**Where:** On page 9 lines 207-213.

**Comment 2:**

Figure 1. "Wind farm layout showing the LiDAR system and the designated study sections for southwest wind conditions. Gray dots represent other wind farms within the same cluster". However, the figure includes two shades of gray dots—light and dark—and

there is no explanation provided in the figure caption or the accompanying text regarding their distinct meanings.

**Response:**

We revised the Fig. 1 caption to clarify all symbol meanings: study turbines are color-coded by section; Dark gray dots change to black dots and indicate turbines within the same wind farm that were excluded from the analysis; and light gray dots indicate turbines in neighboring farms shown for context.

**Where:** on page 1 Fig. 1 caption.

**Comment 3:**

Figure 3 "The raw dataset is shown: green indicates the accepted data used in this study, and red denotes the rejected data. The region II is referring to torque control region and region III to pitch control". However, the actual figure does not contain these colors, which could lead to confusion for readers attempting to interpret the data.

**Response:**

We corrected the Fig. 3 caption to match the figure: blue = accepted, orange = rejected. Also, the figure is now cited in for HDBSCAN filtering.

**Where:** on page 6 Fig. 3 caption, .

**Comment 4:**

Figures 15, 16, and 17 mention " ...before and after applying corrections...". However, the figures currently only display three lines representing TI, shear, and veer without clearly differentiating the data before and after the application of corrections.

**Response:**

We made the before/after comparison explicit in two ways. First, we present paired correlation panels, uncorrected vs. corrected, in the figure pairs Figs. 4-5 (shear), 6-7 (veer), and 8-9 (TI). Second, §4.4 defines the absolute correlation shift that is now explicitly stated and its interpretation is mentioned in the text. We aligned the captions and title of the y-axis of the figures so that their interpretation is clearer. We also made the captions clearer on figures 4,5,6 and 7

**Where:** on page 15 lines 341-344 and page 16 captions of Figures 15,16 and 17, also on page 11 and 12 captions of Figures 4,5,6 and 7

**Comment 5:**

It needs to clearly define how "free-stream TI" and "free stream TI" are calculated.

**Response:**

We now clarified that in this study *free-stream TI* is considered the *front section's* nacelle–SCADA TI, we also harmonize the naming to free-stream in several parts of the manuscript.

**Where:** on page 7 lines 150-151.

**Comment 6:**

In the conclusion section, it mentions "our study underscores the limitations of current flow characterization methods to accurately assess power performance in large-scale offshore wind farms", however, there is limited discussion on "current flow characterization methods" in the introduction section.

**Response:**

We now clarify that the reference is specifically to the IEC framework by revising the conclusion wording to "current *IEC* flow characterization methods".

**Where:** on page 18 line 395.

**Reviewer 2**

"The manuscript entitled Impact of inflow conditions and turbine placement on the performance of offshore wind turbines exceeding 7 MW" deals with the correlation between some parameters of the inflow conditions (veer, shear, turbulence intensity) and the power production at some sections of a wind farm. Some corrections as proposed by the IEC standards are applied and their impact on the results are discussed.

Unfortunately, (1) the poor description of the operational database, (2) the absence of proof of validity of the representativeness (as reference inflow measurement) of the nacelle lidar data located at a big distance from the wind farm of interest and, (3) the very low level of correlations (Pearson coefficient mainly lower than 0.3) between the inflow parameters and the power performance make the present manuscript not suitable for publication." Thank you for your thorough feedback. Your comments prompted us to perform a representativeness study, contextualize our correlation results, and substantially clarify our experimental setup. We believe that these revisions add significant confidence and credibility to our study. In particular:

**Comment 1:**

LiDAR data measurement : (1) No detail on the measurement. Since the lidar is nacelle-mounted, one can assume that the measurement location is following the wind turbine orientation and so, is not always located at the same position. At which distance from the wind turbine is the measurement location? Is it out of the wind farm induction zone? How can you prove this?. (2) The inflow parameters are measured at 8km from the wind farm of interest. How do you prove that these measurements are representative of the inflow impacting the wind farm of interest? Considering the density of wind farms, there are probably strong blockage effects

**Response:**

**(1)** We added a detailed description of the nacelle-mounted lidar in Section 2.1 and Appendix A, covering its scanning mode, averaging time, and the exact upstream range used to derive shear and veer. Because the lidar line-of-sight rotates with the nacelle, the probe-volume position shifts with wind direction; we mitigate this by restricting the analysis to a narrow wind sector, so the probe volume moves much less. The rotation radius is 420 m, which is substantially smaller than the distance from the lidar optical head base to the front section of the farm. This allows us to use the representativeness we added in Appendix B, which indicates that in the offshore North Sea environment, shear and veer are largely invariant over long distances and reduce the concern of the influence due to the nacelle rotation.

Regarding induction, the measurements are outside the *turbine* scale induction zone of the instrumented turbine (we now explicitly report a distance of 2 rotor diameters). Farm-scale induction (the wind-farm induction zone) cannot be ruled out a priori; therefore, in the opening of Section 4 we explain why any farm-scale induction is unlikely to bias our conclusions.

**Where:** on pages 3-4 lines 83-88 and on page 10 lines 264-270, also on the Appendixes A and B.

**(2)** To demonstrate that remote measurements are representative of the inflow affecting the target wind farm, we (i) restrict the analysis to upwind directional sectors that ensure unobstructed retraction to the farm, (ii) compare concurrent statistics between the remote site, the LiDAR nacelle inflow and the coordinates of the front section of the farm, and (iii) perform a multi-year consistency check using the NORA3 North Sea reanalysis data set. The added analysis shows coherent variability and stable shear and veer characteristics between the two locations for the sectors studied. Regarding the possible strong effects, we agree that due to the density of the farms in the area, it is highly probable; however, a full quantification of regional blockage and farm cluster is out of the scope of the present paper; we now state this explicitly and suggest it as future work.

**Where:** on page 4 lines 114-116 and on page 5 lines 120-128 and on page 10 lines 264-270, also Appendix B.

**Comment 2:**

Turbine operational regime: were the 10-min periods used when not all wind turbines were under operation during the 10 minutes? If yes, it means that the level of interactions might be modified by the number of running wind turbines in the wind farm

**Response:**

We use only intervals of 'normal operation', based on SCADA status codes, and explicitly require that all wind turbines on the farm are operating in the same 10-minute period. We explain this now in more detail in the filter section 2.1 an in particular in (Turbine operational regime and environmental dynamics).

**Where:** on page 4 lines 94-95

**Comment 3:**

Wind sector selection: it is 105° wide. The wake interactions are very different depending on the wind direction. This wind sector range is too wide.

**Response:**

We acknowledge that the sector is wide (105°) and includes directions with different wake interactions. This choice is deliberate in order to characterize the *sector-average* behavior (including wake effects) while preserving robust counts per wind-speed bin. Using very narrow sectors can inflate azimuth-bounded phenomena (e.g., half-wake conditions) and produce unstable estimates. Our front/mid/rear grouping intentionally averages across the sector, focusing on each section's mean response to *sector-averaged inflow (including wake effects)* rather than narrow-direction behavior.

**Where:** on pages 4 and 5 lines 113-119.

**Comment 4:**

Figure 3 is not cited in the manuscript. It is not clear how the rejected data had been determined.

**Response:**

Fig. 3 is now cited in the end of the data filtering section and details on the filtering is now

added in the "Outliers (HDBSCAN)" subsection part of the data filtering section.

**Where:** on page 4 lines 105-112 and page 5 line 130.

**Comment 5:**

4 to 9: the Pearson Correlation coefficients values are mainly between 0.3 and -0.3. This magnitude is generally considered as a proof of no correlation between two data sets. One would consider that the interpretation of the evolution of the Pearson coefficient with the wind speed, or the comparison between wind farm segments is fully invalid. If it is not the case, the authors need to explain why such low levels of correlations can be interpreted.

**Response:**

In results and discussion section we added context for why modest pairwise coefficients are expected in operational data. We explain now that the interpretation of the results does not rely on large absolute magnitudes; instead, we focus on pattern differences and sign changes between section, operational regime dependence with wind speed, and pre/post-correction trends differences. We also refer to a similar study that reported similarly low coefficients under comparable conditions.

**Where:** on page 10 lines 255-263

**Comment 6:**

2 and Line 91: 50% or 64% of the data are gathered in this wind sector range?

**Response:**

We correct the share to **64%** across text and wind-rose caption.

**Where:** on page 5 Fig. 2 caption, on page 4 line 113.

**Comment 7:**

Lines 87–88: More details on the DB-scan algorithm.

**Response:**

We document the outlier removal step in *Outliers (HDBSCAN)* section, stating features and parameter choices.

**Where:** on page 4 lines 105-112.

**Comment 8:**

Fig 3: "blue" instead of "green".

**Response:**

Caption corrected to **blue = accepted**, **orange = rejected**, and the figure is now cited where the filtering is described.

**Where:** on page 6 Fig. 3 caption.

**Comment 9:**

3.4.2 Turbulence intensity correction and Equation 3: it is not clear what the "simulated average power" refers to. Which simulation?

**Response:**

We define now on text the $P_{\mathrm{sim}}$ and provide more details for the turbulence intensity correction.

**Where:** on pages 8-9 lines 205-211.

**Comment 10:**

Lines 167–168: explain the Sliding Window Pearson Correlation approach.

**Response:**

We added a concise, description of the sliding–window Pearson correlation method and added details on the parameters we used in section 3.5.

**Where:** on page 9 lines 213-233

**Comment 11:**

7 is not cited in the manuscript.

**Response:**

We now cite Fig. 7 together with Fig. 6 in the veer discussion to guide the before/after comparison.

**Where:** on page 11 line 288.

**Reviewer 3**

We are grateful for your insightful comments, which challenged us to refine our central hypothesis, better contextualize the study's contribution, and overall create a stronger manuscript. Specifically, your feedback prompted us to conduct a representativeness study, add a brief discussion on the uncertainty of our lidar measurements, and better contextualize our correlation results and expand our discussion section, additions that we believe have led to a more impactful and defensible manuscript.

**Comment 1:**

This manuscript investigates the correlation between atmospheric inflow parameters—namely wind shear, veer, and turbulence intensity—and offshore wind turbine power production. The analysis is framed in the context of IEC-recommended normalization, with the goal of exploring performance dependencies across wind speed bins. The topic is clearly relevant for wind energy research and operations, particularly as offshore turbines exceed 7 MW in rated capacity and sit within more variable marine boundary layer conditions.

However, it is not clear what novel contribution this manuscript makes to the field or the central hypothesis being investigated. IEC normalization approaches are not intended to serve as predictive models, but rather to establish standard design and compliance conditions. The fact that these normalized parameters correlate weakly with power production is not necessarily surprising, and the authors need to contextualize their work more rigorously. If the intent of the paper is to show that IEC-style normalization underrepresents atmospheric variability or leads to weak statistical connections to power, that point must be made explicit and defended with clear analysis. Even better would be to outline a meaningful improvement to the approach outlined in the IEC Standards.

In its present form, the paper suffers from several methodological and framing issues that limit its scientific clarity. Many of the assumptions are not justified, key steps in the analysis are missing or difficult to understand, and several potentially confounding factors are not adequately discussed.

**Response:**

We believe this concern is now addressed because the revised manuscript (i) makes the *central hypothesis* explicit—*evaluate* IEC 61400-12-1 flow characterization (REWS, TI) *inside* wake affected regions of a farm, where performance may differ from free-stream; (ii) makes the hypothesis testable by dividing the wind farm into sections (front / middle / rear) and comparing uncorrected vs. IEC-corrected results; and (iii) quantifies effects using $\Delta|\rho|$ (net change in coupling), supported by the sensitivity coefficient $\beta$ and MAD analyses for interpretability. The results show a limited benefit for shear/veer and TI overcorrection in rear rows, which we declare clearly in the conclusion as a limitation of current IEC flow characterization methods in wakes, with concrete next steps (e.g. TKE). In summary, we clarify the aim and provide a targeted analysis to test it.

**Where:** on page 2 lines 45-48 and on page 3 lines .

**Comment 2:**

The nacelle-mounted lidar appears to be located much farther from the wind plant than indicated in the text. Figure 1 suggests a distance closer to 18–20 km rather than the 8 km noted. This large separation raises serious concerns about the representativeness of the inflow measurements used in the correlation analysis. The authors should justify the use of this lidar system for characterizing inflow to the plant or clearly state the limitations that this introduces.

**Response:**

We apologize for our mistake and we corrected the separation to $\sim 23.5\,\mathrm{km}$. To support representativeness for that large distances, we (i) restrict to unobstructed upwind sectors, (ii) add a homogeneity study using NORA3 dataset. The Nora3 study in the Appendix B showed a coherent shear and veer statistics between the lidar position and the farm front section over the studied sector. We also explicitly acknowledge the limitations due to the separation distance.

**Where:** on pages 4 and 5 lines 113-119, on page 5 lines 120-128 and on the Appendix B.

**Comment 3:**

The manuscript lacks sufficient discussion of existing literature on nacelle-mounted lidar systems and their associated uncertainties, especially for continuous-wave systems used at long range. Important work should be cited and discussed to help the reader assess the quality and validity of the measurement inputs. Letizia et al., 2021, is a good place to begin.

**Response:**

We added a short uncertainty methods note for CW nacelle-lidar and cite relevant studies. We also added details on the exact configuration of the lidar during the time of the study and the estimation of the shear and veer.

**Where:** on page 3 lines 83-88, page 7 lines 166-173 and Appendix A.

**Comment 4:**

The authors assume that wind shear and veer estimates from the distant lidar are representative of conditions across the entire wind plant. This is a strong assumption, especially in the offshore context, where wind direction and shear can vary significantly across distances of several kilometers. The authors should validate this assumption using available SCADA data, such as nacelle wind speed or yaw measurements, or else explicitly acknowledge and try to quantify the uncertainty introduced by this assumption.

**Response:**

We now (i) state the representativeness assumption explicitly; and (ii) compare concurrent distributions of shear and veer between the upwind lidar and the wind farm's front section using NORA3 reanalysis to assess spatial gradients across the separation (Appendix B). These checks indicate broadly similar central tendencies and variability at the two locations, increasing confidence that the upwind measurements are representative of the *front-section inflow* used in our analysis.

Crucially, we do not assume that shear and veer remain unchanged within the array. On the contrary, their wake-modified evolution across rows is part of what our methodology is

attempted to quantify (via the front/mid/rear sectioning and sliding-window analyses). We therefore treat the upwind lidar as providing boundary conditions at the farm entrance, and we analyze how coupling and correction performance evolve downstream. Residual spatial heterogeneity remains listed as a limitation.

**Where:** page 7 lines 166-173.

**Comment 5:**

The data selection process is not fully described. Specifically, it is not clear whether or how dynamic atmospheric events (e.g., wind speed ramps, gusts, frontal passages, etc.) are filtered out of the data set. These events can have a strong impact on the correlation strength between inflow and power production. The authors should consider filtering based on time series dynamics (e.g., Hamilton, 2020) to isolate quasi-stationary atmospheric periods. This is likely to produce stronger correlations between observed quantities.

**Response:**

We have clarified the filtering criteria and operational-status requirements in the manuscript. Dynamic atmospheric events (ramps, gusts, frontal passages) are intentionally retained to reflect realistic, farm-wide operating conditions. Consequently, the reported correlations should be read as conservative rather than inflated by strict stationarity screening. Outlier handling and additional screening steps are now described in detail. The reasoning behind this choice is that while isolating quasi-stationary periods could produce higher correlations, it would remove variability that turbines routinely experience.

**Where:** on page 4 lines 95-112.

**Comment 6:**

In Figure 1, several turbines within the wind plant are shown in gray and appear to be excluded from the analysis, but the reason for this exclusion is not stated. Similarly, if there are turbines located closer to the source of atmospheric measurements, perhaps they should be considered instead.

**Response:**

We revised the caption and text: colored markers denote study turbines (front/middle/rear). The reasoning behind the exclusion of some wind turbines within the wind farm is now discussed in the selection division. Regarding the use of data of wind turbines closer to the source of atmospheric measurements unfortunately the data of these wind farms were not available to us or their size was not adequate for this study.

**Where:** on page 7 lines 155-156.

**Comment 7:**

The paper mentions the use of DB-SCAN for outlier detection but provides no mathematical description or reference. At a minimum, a citation should be added, along with a brief description of how the DB-SCAN parameters were selected. Moreover, it appears that the DB-SCAN filtering may bias the power deviation metric discussed in Section 3.2. By excluding data points that deviate from the nominal power curve, the authors may artificially reduce the observed variability. This potential bias should be acknowledged and discussed.

**Response:**

We clarified that we use *HDBSCAN* and documented the features, the distance metric, and key parameters; we also now cite the method used in this study and state the limitation due to artificially reduced variability.

**Where:** on page 4 lines 106-113.

**Comment 8:**

The analysis sector from 180° to 285° includes turbines that are likely to be in the wake of upwind turbines. The authors should discuss the extent to which wake effects influence turbulence intensity estimates and correlation strength. It may be appropriate to apply a wake-added turbulence model or include a wake filter to avoid spurious correlations driven by wake-induced variability.

**Response:**

Our aim is to evaluate IEC corrections within a wind farm, so we intentionally retain wakes rather than filter or model them away. Wake exposure is controlled by design via a front / middle / rear partition (free stream progressively waked), and we evaluated both absolute correlations and pre/post corrections ($\Delta|\rho|$). This framing directly reveals that TI behaves differently downstream and that the IEC TI correction can over-compensate in the rear at low speeds—effects that would be obscured by wake removal. To explain this, we added a sentence in the methods clarifying this scope. We therefore keep wakes in the baseline while acknowledging wake-added TI as a mechanism underlying the observed row dependence.

**Where:** on page 4 lines 114-120 and page 8 lines 196-200.

**Comment 9:**

The correlation analysis itself is not clearly explained. The use of a 0.75 m/s sliding window appears central to the results, but it is not described in sufficient detail. Does the window represent a binning interval for mean values, a pairwise filter on measurement differences, or something else? How are variables with different units (e.g., shear, veer, TI) handled in this scheme? Further, what range of wind speeds is included in the analysis, and how is the resolution of the sliding window chosen? The authors may wish to define the correlation structure in terms of normalized wind speed (e.g., $U/U_{rated}$) to provide more generalizable insight.

**Response:**

We clarified how the sliding-window Pearson correlation is constructed and used. Records are ordered by wind speed, correlations between each environmental variable and power deviation are evaluated within overlapping windows along this axis, and results are presented against normalized wind speed for generality. We explain the motivation for the windowing approach, the wind-speed range considered (covering torque- and pitch-control regimes), and the parameters and details for applying the window. We also note that the correlation metric is unitless, so differently scaled variables are handled consistently. All the correlations are defined in terms of normalized wind speed. Unfortunately, we cannot disclose the exact wind speed range for confidentiality reasons.

**Where:** on page 9 lines 215-235.

**Comment 10:**

Figures 4–11 present a variety of correlations between atmospheric parameters and power

output or power deviation, but the organization of these figures is unclear. The alternation between absolute power and deviation metrics should be explained more explicitly. In most cases, the correlation coefficients are below 0.4. Although low correlations are not inherently uninteresting, the authors should explore and discuss the potential reasons for these weak results. For instance, are the variables themselves weakly coupled, is the inflow poorly characterized, or are key intermediate variables missing from the analysis? The paper would benefit from a thoughtful discussion of these points, which could help inform future efforts to better predict wind turbine performance from inflow metrics.

**Response:**

We clarified the figure logic add the definition of the metric $\Delta|\rho| \equiv |\rho_{\text{corrected}}| - |\rho_{\text{uncorrected}}|$, and now use consistent axis labels and legends that explicitly indicate when correlations are calculated with the power deviation (PD). We also added a brief paragraph in the Results and discussion section, in which we contextualize our results by explaining why pairwise correlations are expected to be modest in operational data, while we also include citations to a similar offshore empirical study with similar low correlations.

**Where:** on page 10 lines 255-263 and section. 4.4.

**Global revision due to representativeness update.** Following the representativeness assessment (App. B) and we implement the additional filter for extreme shear and veer and we performed the calculations based on the new inflow. The results of TI changed only marginally. However, the magnitudes of shear and veer and their SWC profiles changed sufficiently to warrant updates to the corresponding figures and text in Sec. 4.4 Our primary conclusions remain unchanged. The refined inflow conditions make the row- and section-dependent patterns clearer. All figures in the Data and Results sections that depend on inflow filtering were regenerated with the new data, and the most noticeable differences occur in Figs. 15, 16 and 17.

**Where:** on pages 16 and 17 of section 4.4, on Figures 3-17